# Associating Inulin with a Pea Protein Improves Fast-Twitch Skeletal Muscle Mass and Muscle Mitochondrial Activities in Old Rats

**DOI:** 10.3390/nu15173766

**Published:** 2023-08-28

**Authors:** Jérôme Salles, Marine Gueugneau, Véronique Patrac, Carmen Malnero-Fernandez, Christelle Guillet, Olivier Le Bacquer, Christophe Giraudet, Phelipe Sanchez, Marie-Laure Collin, Julien Hermet, Corinne Pouyet, Yves Boirie, Heidi Jacobs, Stéphane Walrand

**Affiliations:** 1Unité de Nutrition Humaine (UNH), Université Clermont Auvergne, INRAE, CRNH Auvergne, 63000 Clermont-Ferrand, France; marine.gueugneau@inrae.fr (M.G.); veronique.patrac@inrae.fr (V.P.); christelle.guillet@uca.fr (C.G.); olivier.le-bacquer@inrae.fr (O.L.B.); christophe.giraudet@inrae.fr (C.G.); phelipe.sanchez@uca.fr (P.S.); m-laure.collin@uca.fr (M.-L.C.); julien.hermet@inrae.fr (J.H.); corinne.pouyet@inrae.fr (C.P.); yves.boirie@inrae.fr (Y.B.); stephane.walrand@inrae.fr (S.W.); 2Cosucra-Groupe Warcoing S.A., 7740 Warcoing, Belgium; cmalnero-fernandez@cosucra.com (C.M.-F.); hjacobs@cosucra.com (H.J.); 3Unité de Nutrition Humaine (UNH), Université Clermont Auvergne, INRAE, PlateForme d’Exploration du Métabolisme, MetaboHUB-Clermont, 63000 Clermont-Ferrand, France; 4CHU Clermont-Ferrand, Service Nutrition Clinique, 63000 Clermont-Ferrand, France

**Keywords:** inulin, pea protein, sarcopenia, skeletal muscle, protein synthesis, mitochondrial activity

## Abstract

Aging is associated with a decline in muscle mass and function, leading to increased risk for mobility limitations and frailty. Dietary interventions incorporating specific nutrients, such as pea proteins or inulin, have shown promise in attenuating age-related muscle loss. This study aimed to investigate the effect of pea proteins given with inulin on skeletal muscle in old rats. Old male rats (20 months old) were randomly assigned to one of two diet groups for 16 weeks: a ‘PEA’ group receiving a pea-protein-based diet, or a ‘PEA + INU’ group receiving the same pea protein-based diet supplemented with inulin. Both groups showed significant postprandial stimulation of muscle p70 S6 kinase phosphorylation rate after consumption of pea proteins. However, the PEA + INU rats showed significant preservation of muscle mass with time together with decreased MuRF1 transcript levels. In addition, inulin specifically increased PGC1-α expression and key mitochondrial enzyme activities in the plantaris muscle of the old rats. These findings suggest that dietary supplementation with pea proteins in combination with inulin has the potential to attenuate age-related muscle loss. Further research is warranted to explore the underlying mechanisms and determine the optimal dosage and duration of intervention for potential translation to human studies.

## 1. Introduction

As the global population continues to grow, there is increasing pressure on environment and food resources. The production of animal-based protein, such as beef and lamb, is associated with significant environmental challenges [1]. Plant-based agriculture uses less natural resources such as water and land, generates fewer greenhouse gas emissions, has a lower carbon footprint, and helps preserve biodiversity [2]. By incorporating more plant protein into the food system, we can mitigate environmental degradation and work towards a more sustainable future. Plant protein also diversifies the food supply and contributes to global food security. Growing plant-based proteins is less resource-intensive and more adaptable to different regions, making it accessible to a broader range of communities [3].

Plant-based protein sources are rich in certain essential amino acids, vitamins, minerals, and dietary fiber, making them a valuable source of nutrition. They contribute to a well-balanced diet and support various bodily functions, including muscle growth and repair, and overall health [4]. Higher consumption of plant-based protein sources is associated with several health benefits, including reduced risk for chronic diseases such as heart disease, type 2 diabetes, and certain types of cancer [5]. Plant-based diets have been linked to lower cholesterol levels, improved weight management, and better overall health outcomes [5].

Sarcopenia refers to the age-related loss of skeletal muscle mass, strength, and function. The condition is characterized by a progressive decline in skeletal muscle mass and function, which can lead to decreased mobility, increased risk of falls and fractures, and a decline in overall physical performance. Loss of muscle tissue is a common condition that affects many older adults, typically starting around the age of 40 and becoming more pronounced after the age of 65. Although sarcopenia is a natural part of the aging process, it can be accelerated by certain factors, such as physical inactivity, poor nutrition, hormonal changes, and chronic diseases [6,7]. Symptoms of sarcopenia include muscle weakness, reduced stamina and resistance, difficulty performing everyday tasks, and changes in body composition (increased body fat and decreased muscle mass). However, these symptoms vary in severity among individuals [8]. The causes of sarcopenia cover in particular a deregulation of muscle protein metabolism and an alteration of muscle mitochondrial function.

Preventing and managing aging-related loss of skeletal muscle demands a multi-faceted approach that includes regular exercise, proper nutrition, and lifestyle modifications. Among these, adequate protein intake is essential for muscle maintenance in older subjects, mainly through the regulation of muscle synthesis rate [6]. Plant proteins can play a valuable role in managing sarcopenia, especially when incorporated into a well-balanced diet that meets individual nutritional needs. While animal-based protein sources have traditionally been considered the gold standard for muscle health, plant-source proteins can provide a variety of essential amino acids necessary for muscle synthesis and maintenance. For instance, legumes such as peas, beans, lentils and chickpeas are valuable sources of plant proteins [4], especially when provided with grain protein to balance the amino acid profile. We previously demonstrated that net protein utilization in old rats fed only legume-enriched pasta diets was similar to those fed a casein diet [9]. The legume-enriched pasta diet was found to induce a similar muscle protein synthesis rate and an equivalent effect on muscle weight and muscle protein accretion in old rats compared to a casein diet [9]. Blending wheat and legumes in a food staple, such as pasta, improved its essential amino acid profile by making it more adequate for muscle synthesis rate and muscle protein accretion, especially for older individuals. Sources of plant proteins also offer dietary fiber, vitamins, minerals, and antioxidants that all support overall health. Studies have shown that the consumption of some of these nutrients makes it possible to optimize the effectiveness of dietary proteins on the regulation of muscle protein metabolism. For example, in old rats, regular antioxidant intake increased the effect of protein intake on muscle protein synthesis rate [10]. This same action has also been demonstrated for other micronutrients, such as vitamin D [11,12,13].

Inulin is composed of chains of fructose molecules linked together and belongs to a group of carbohydrates known as fructans. Inulin is not digestible by human digestive enzymes, which means it reaches the colon intact, where it can serve as a prebiotic. Beneficial gut bacteria, such as bifidobacteria and lactobacilli, can ferment inulin, producing short-chain fatty acids (SCFA) as by-products, which have several health benefits for the human body, such as anti-inflammatory effects or blood glucose regulation. A recent study has shown that the prebiotic 1-kestose drove recovery from muscle atrophy in older patients with sarcopenia [14]. 1-kestose is the inulin precursor naturally present in various fruits and vegetables and it can be readily extracted from many plants [15]. All patients administered 1-kestose for 12 weeks showed an increase in skeletal muscle mass index and a decrease in body fat percentage. The study went on to explore the gut microbiota and found that the *Bifidobacterium longum* population was significantly increased in the intestine after 1-kestose administration [14]. However, the change in gut microbial composition was subtle and transient, so further studies are needed in order to firmly conclude that prebiotics have a beneficial effect on muscle mass and function, and to define the mechanisms underlying this action.

Here we hypothesized that consumption of a prebiotic such as inulin in addition to a plant-based diet rich in pea protein would increase the effect of the plant protein on muscle mass in old rats and that this effect would be associated with profound changes in muscle protein and energy metabolism. We thus designed this study to establish the links between consumption of a pea protein diet with or without dietary fiber, i.e., inulin, on muscle protein turnover and mitochondrial activity in old rats.

## 2. Materials and Methods

### 2.1. Animal Experiment

Animal procedures were performed in accordance with the European guideline on the care and use of laboratory animals (2010-63UE). Experimental protocols were reviewed and approved by the local institutional animal care and use committee (authorization number: APAFIS#5329-2016051115541284 v2). Animals were housed in the INRAE’s Human Nutrition Research animal facility (agreement No. E6334515).

A total of forty 20-month-old male Wistar rats were purchased from the Janvier Labs breeding facility (Le Genest-St-Isle, France). All animals came from the same batch, and were bred under the same conditions throughout their lives. The rats were individually housed in plastic cages and maintained at 22 °C under a 12 h dark–12 h light cycle with free access to water, as previously described [16]. All the rats were fed a maintenance diet (A04, Safe, Augy, France) ad libitum for two weeks. After this adaptation period, the rats were randomized into two groups according to body weight, fat mass and lean mass. Animals were assigned (n = 20 per group) to a diet containing 14% pea protein (PEA rats), i.e., PISANE™ C9 (Cosucra, Warcoing, Belgium) or a diet containing 14% pea protein supplemented with 7.5% inulin (FIBRULINE™ Instant, Cosucra, Warcoing, Belgium) (PEA + INU rats) for 16 weeks. Diets were produced by the “Sciences de l’Animal et de l’Aliment de Jouy” (SAAJ) experimental unit (INRAE, Jouy en Josas, France) (Table 1). Body weight and food intake were measured weekly. At the end of the experiment, the remaining rats (PEA rats; n = 14 and PEA + INU; rats n = 12) were fasted overnight but with free access to water. Each rat group was then randomly divided into two subgroups that either were kept in the fasted state or given a nutrient mixture by oral gavage as a nutrient stimulus (postprandial state). The nutrient mixture contained amino acids, sucrose and glucose (see Table 2 for composition data) and was administered at a volume corresponding to 1 mL/100 g body weight [17,18,19]. The nutrient bolus contained ≈33% of the daily amount of leucine ingested by both rat groups (PEA and PEA + INU rats) during the experimental protocol to stimulate muscle protein synthesis, and carbohydrates to supply energy and induce acute insulin secretion. Fasted rats and postprandial-state rats were anesthetized by isoflurane inhalation (60 min after oral gavage for postprandial-state rats). Blood samples were collected from the abdominal aorta, and plasma was prepared by centrifugation. Liver, heart, adipose tissues and hindlimb skeletal muscles were removed, weighed, snap-frozen in liquid nitrogen, and stored at −80 °C for later analysis.

### 2.2. Body Composition Measurement

Body composition was measured to assess the effect of the two diets on the change of lean mass and fat mass of the aged animals. Fat mass and lean body mass (g) were determined on live non-anesthetized rats using an EchoMRI-100 body composition analyzer (Echo Medical Systems LLC, Houston, TX, USA).

### 2.3. Protein Quality Evaluation

Dietary quality of proteins can be defined by the nitrogen balance, i.e., an estimate of body nitrogen retention. In the last week of the experiment, rats were placed in metabolic cages for the last 4 days. Urine and feces were collected, and total excreted nitrogen was determined by at Institut UniLaSalle (Beauvais, France) by the Dumas method [20]. Nitrogen balance (NB) was calculated using the equation [21].

$$NB\left(g\right)=NI-\left(FN+UN\right)$$

where NI is nitrogen intake, FN is fecal nitrogen, and UN is urinary nitrogen. Fecal and urinary endogenous nitrogen excretion were deduced from a group of old rats that received a nitrogen-free diet during the metabolic cage period [16].

### 2.4. Plasma Analyses

The overall metabolic effects of the two diets were assessed by determining blood tests. Plasma levels of glucose and triglycerides were determined using a Konelab 20 analyzer (Thermo-Electron Corporation, Waltham, MA, USA). ELISA kits (Alpco Diagnostics, Salem, NH, USA) were used to determine plasma insulin levels in both fasted rats and postprandial-state rats.

### 2.5. Protein Synthesis Measurement

One of the dynamic markers of muscle protein turnover is the evaluation of the rate of protein synthesis using isotopic tracers. Muscle protein synthesis was studied by measuring the rate of incorporation of a stable isotope, i.e., an AA L-[^13^C_6_]-labeled phenylalanine (Eurisotop Saint-Aubin, France), into muscle proteins using the flooding dose method. Fasted rats and postprandial rats (at 10 min after oral gavage) were injected subcutaneously with a large dose of L-[^13^C_6_] phenylalanine (50% mol excess, 150 µmol/100 g) to flood the precursor pool for protein synthesis. Tracer incorporation time was 50 min in both groups. A 50-mg piece of plantaris muscle was used to isolate and hydrolyze total mixed proteins as previously described [9,16,22,23,24]. After derivatization, L-[^13^C_6_] phenylalanine enrichments in hydrolyzed proteins and in tissue fluid were assessed using gas chromatography–mass spectrometry (Hewlett-Packard 5971A; Hewlett-Packard Co., Palo Alto, CA, USA). The absolute synthesis rate (ASR) of proteins was calculated using the equation:
$$ASR=\frac{Ei}{Ep\times t}\times 100\times TPC$$

where Ei is enrichment as atom percent excess of L-[^13^C_6_] phenylalanine derived from phenylalanine from proteins at time t (minus basal enrichment), Ep is mean enrichment in the precursor pool (tissue fluid L-[^13^C_6_] phenylalanine), t is incorporation time in hours, and TPC is the total protein content in mg. ASR data were expressed as milligrams of proteins per hour (mg/h).

### 2.6. Quantitative RT-PCR Analysis

Some muscle metabolic markers altered by dietary protein quality have been determined by measuring their transcript level. This is the case for markers of muscle proteolysis and mitochondrial biogenesis. The protocol used here for total RNA extraction and mRNA analysis has been detailed elsewhere [16,23,25]. Briefly, total RNA was extracted from a piece of frozen plantaris or perirenal adipose tissue using Tri-Reagent according to the manufacturer’s instructions (Euromedex, Mundolsheim, France). RNA was quantified by spectrophotometry at 260 nm. Concentrations of mRNAs corresponding to genes of interest were measured by reverse transcription followed by PCR amplification. Total RNA was reverse-transcribed using SuperScript III reverse transcriptase and a combination of random hexamer and oligo dT primers (Invitrogen, Life Technologies, Saint-Aubin, France). PCR amplification was performed using 2× Rotor-Gene SYBR Green PCR master mix and a Rotor-Gene Q system (Qiagen, Courtaboeuf, France). Relative mRNA concentrations were analyzed using Rotor-Gene software (version 2.3.1) and the linear standard curve method. Table 3 lists the primers used for real-time PCR amplification. mRNA contents were normalized to hypoxanthine-guanine phosphoribosyltransferase (HPRT) level. Data were expressed in arbitrary units.

### 2.7. Western Blot Analysis

To complete the evaluation of muscle protein synthesis, we evaluated the activation state of its main regulatory pathway, the p70 S6 kinase pathway. Plantaris muscles were homogenized in an ice-cold lysis buffer (50 mM HEPES pH 7.4, 150 mM NaCl, 10 mM EDTA, 10 mM NaPPi, 25 mM β-glycerophosphate, 100 mM NaF, 2 mM Na orthovanadate, 10% glycerol, 1% Triton X-100) containing a protease inhibitor cocktail (1%) as previously described [12,25,26]. Denatured proteins were separated by SDS-PAGE on a polyacrylamide gel and transferred to a polyvinylidene membrane (Millipore, Molsheim, France). Immunoblots were incubated in a blocking buffer and then probed with anti-phospho p70 S6 kinase (Thr389) and anti-total p70 S6 kinase primary antibodies (Cell Signaling Technology, Ozyme, Saint-Quentin-en-Yvelines, France). The immunoblots were then incubated with horseradish peroxidase-conjugated swine anti-rabbit immunoglobulins (DAKO, Trappes, France). Luminescent visualization was done using ECL Western Blotting Substrate (Pierce, Thermo Fisher Scientific, Courtaboeuf, France) and a Fusion Fx imaging system (Vilber Lourmat, Collegien, France). Density of the bands was quantified using MultiGauge 3.2 software (Fujifilm Corporation). The phosphorylation state of p70-S6 kinase in the plantaris muscle exhibited a strong difference between the fasting and postprandial states. Consequently, samples from fasted and postprandial rats were separated onto individual gels, while a consistent mixture of all samples was loaded on each gel to facilitate comparison of the blots. The values represent the ratio of the phosphorylated protein levels to total protein levels, and are expressed in arbitrary units.

### 2.8. Mitochondrial Enzymatic Assays

Since dietary protein, especially plant-source protein, can change mitochondrial activity [16], we assessed mitochondrial function. Fifty mg of frozen rat plantaris muscles was homogenized in a glass–glass Potter in 9 volumes of homogenization buffer (225 mM mannitol, 75 mM sucrose, 10 mM Tris-HCl, 10 mM EDTA, pH 7.2) and spun down at 650× *g* for 20 min at 4 °C. The supernatant was kept, and the pellet was suspended in 9 volumes of homogenization buffer and re-submitted to the same spin-down procedure. Both supernatants were pooled and used for the assay. After protein quantification, activities of complex 1, citrate synthase and 3-hydroxyacyl-CoA dehydrogenase (HAD) were spectrophotometrically assayed as previously described [23,25,27,28]. Complex I and HAD activities were spectrophotometrically assayed in the supernatant fraction by following the oxidation of nicotinamide adenine dinucleotide, reduced form (NADH). Citrate synthase activity was measured by following the reduction of 5,5-dithiobis(2-nitrobenzoic acid) (DTNB) [23,25,27,28,29]. All activities were expressed as fold change vs. the value found for the PEA group.

### 2.9. Statistics

All data were expressed as means ± SEM. Animals that died or developed tumors during the experiment were excluded from the analysis. In detail, while we had 20 rats per group at baseline, the number of rats remaining at the end of the experiment was 14 PEA rats and 12 PEA + INU rats. The data were analyzed for homogeneity of variance and normality. Homogeneous data were analyzed using an unpaired *t*-test. Heterogeneous data were analyzed using the non-parametric Mann–Whitney U test. To compare the initial values of body weight, fat mass and lean mass with the final values, data were analyzed by a paired *t*-test. Differences were considered significant at *p* < 0.05. Statistical analysis was performed using NCSS 2020 software (NCSS LLC., Kaysville, UT, USA) and StatView software (version 4.02; Abacus Concepts, Berkeley, CA, USA).

## 3. Results

### 3.1. Body Weight and Composition Changes and Final Relative Tissue Weights

There was no significant between-group difference in food intake over the course of the study (21.8 ± 0.6 g/day and 21.3 ± 0.6 g/day for PEA and PEA + INU rats, respectively). At the beginning and at the end of the study, there was no between-group difference in body weight, fat mass and lean mass. PEA and PEA + INU rats tended to gain body weight, significantly gained fat mass and significantly lost lean mass over the course of the 16-week experimental period, and these changes were similar between groups (Table 4).

Tissue mass-to-body weight ratios were calculated for different lean soft and adipose tissues and reported in Table 5. The relative weights of fast-twitch skeletal muscles, i.e., plantaris, tibialis and gastrocnemius, were significantly higher in PEA + INU rats than in PEA rats (+16%, +16% and +18%, *p* < 0.05, respectively). Conversely, PEA + INU diet tended to decrease the relative weight of perirenal adipose tissue (*p* = 0.06). Note that relative weights of soleus (slow-twitch skeletal muscle), liver, heart and subcutaneous adipose tissue were similar in both groups (Table 5).

### 3.2. Protein Quality Evaluation

Nitrogen intake and fecal and urinary nitrogen contents were measured during the metabolic cage period (Table 6). Nitrogen intake and urinary nitrogen content were similar between the two rat groups whereas fecal excreted nitrogen was significantly higher for PEA + INU rats than PEA rats. Fecal nitrogen content-to-nitrogen intake ratio was significantly higher in PEA + INU rats than in PEA rats (+25%, *p* < 0.05) whereas urinary nitrogen content-to-nitrogen intake tended to be lower in PEA + INU rats than in PEA rats (−16%, *p* = 0.05; Table 6). As a result, the nitrogen balance, which is the difference between nitrogen intake and nitrogen loss by both fecal and urinary routes, was slightly greater in rats fed the PEA + INU diet than in rats fed the PEA diet (*p* = 0.07).

### 3.3. Muscle Protein Synthesis in Fasted and Postprandial State and Markers of Muscle Proteolysis

To investigate the molecular mechanisms involved in the increase of relative weight of fast-twitch skeletal muscles in PEA + INU rats compared to PEA rats, we explored protein synthesis and gene expression of markers of muscle proteolysis in plantaris muscles in fasted state and postprandial state. To mimic the postprandial state, we administered a solution of amino acids and carbohydrates by oral gavage. Oral administration induced a significant increase in plasma glucose levels and, consequently, a significant increase in plasma insulin concentration in both rat groups (Table 7). Note that plasma levels of glucose, insulin and triglycerides were not significantly different in both PEA and PEA + INU rats in both fasted state and postprandial state (Table 7). Analysis of the data for groups pooled together found that the oral gavage with the nutrient mixture enhanced muscle absolute synthesis rate (ASR) in comparison with the fasted state (+42%, *p* < 0.05). However, analysis of the data for individual groups found that oral gavage did not significantly increase muscle ASR compared to the fasted state (Figure 1A), despite a strong activation by phosphorylation of p70 S6 kinase, an intermediate of the translation initiation step, in the postprandial state compared to the fasted state (Figure 1B,C). There was no between-diet (PEA vs. PEA + INU) difference in muscle ASR (Figure 1A) or p70 S6kinase phosphorylation level (Figure 1B,C) in the fasted state or in the postprandial state. We then assessed the involvement of the ubiquitin-proteasome pathway in the regulation of skeletal muscle mass in PEA and PEA + INU rats by measuring transcript levels of MuRF1 and MAFbx. MAFbx mRNA expression was unaffected by both experimental diets and both nutritional states (Figure 1D). MuRF1 transcript levels were similar between diet groups in the fasted state but significantly lower in plantaris muscles of PEA + INU rats compared to PEA rats in the postprandial state (Figure 1E). Oral gavage with the nutrient mixture induced a significant downregulation of muscle MuRF1 mRNA expression in PEA + INU rat muscles but not in PEA rat muscles (Figure 1E).

### 3.4. Muscle Mitochondrial Activity

We investigated the effect of inulin supplementation on muscle mitochondrial function in old rats by measuring the maximal activities of complex 1, which is an electron transport chain complex, citrate synthase, which is a mitochondrial matrix enzyme used as a marker of mitochondrial density, and 3-hydroxyacyl-CoA dehydrogenase (HAD), which is a key enzyme of the mitochondrial β-oxidation pathway. Complex 1, citrate synthase and HAD activity showed no nutritional state-related changes, so we pooled the muscle activity measurements done in fasted state with the muscle activity measurements done in postprandial state for each of the three enzymes. Complex 1 activity tended to be enhanced in plantaris homogenates of PEA + INU rats compared to PEA rats (*p* = 0.07) (Figure 2A). Inulin supplementation induced an increase in citrate synthase and HAD activity in plantaris muscles (+18% and +28% vs. PEA rats, respectively, *p* < 0.01; Figure 2A). To investigate the molecular mechanisms involved in the modulation of muscle mitochondrial function in response to dietary inulin supplementation, we analyzed the gene expression levels of peroxisome proliferator-activated receptor gamma coactivator 1 alpha (PGC1α) to estimate mitochondrial biogenesis in plantaris muscles. In line with the increase in muscle mitochondrial function in inulin-supplemented old rats, plantaris muscles of PEA + INU rats showed a strong increase of PGC1α transcripts compared to PEA rats (+64%, *p* < 0.05; Figure 2B).

## 4. Discussion

This study demonstrates that pea protein combined with inulin improves muscle anabolism and muscle mitochondrial activity, suggesting potential benefits for muscle health in old rats. The weight of fast-twitch muscles, which are precisely the muscles most damaged by aging, was higher in old rats following consumption of a diet based on pea protein plus dietary fiber (inulin) compared with pea protein given without dietary fiber. Mitochondria are the energy powerhouses of cells, including muscle cells, so as pea protein with inulin improves mitochondrial function and efficiency, it could potentially also enhance energy production, which is crucial for optimal muscle function and performance. These findings align with the potential benefits of plant proteins and inulin separately identified in previous specific studies [9,16,30]. Pea protein is a good source of essential amino acids, which are crucial for muscle protein turnover. Inulin, acting as a prebiotic, can positively influence gut microbiota, leading to reduced inflammation and metabolic improvements.

The primary aim of this study was to evaluate the quality and impact of pea protein enriched with inulin on protein retention in old rats compared to pea protein given without inulin. The nutritional value of proteins is dependent on their amino acid composition and how readily they can be digested, absorbed, and incorporated into body proteins [4]. As expected, pea protein and inulin lead to a higher fecal nitrogen excretion in old rats, resulting in a higher ratio of fecal nitrogen-to-nitrogen intake. The presence of the dietary fiber, e.g., inulin, could interfere with protein digestion and increase nitrogen excretion. Dietary fiber has been shown to increase the secretion of nitrogenous substances into the alimentary tract [31,32,33] and to increase the quantity of endogenous amino acids at the terminal ileum of simple-stomached animals [34]. It has also been reported that nitrogenous secretions including pancreatic juice [31,32] and mucus [33] are secreted in larger amounts when experimental animals are fed purified diets supplemented with fiber. Furthermore, Bergner et al. [35] reported that dietary fiber hindered the reabsorption of endogenous amino acids from the small intestine. However, here, body weight and lean mass changes were similar between old rats fed pea protein and pea protein plus inulin, despite the higher fecal nitrogen-to-nitrogen intake ratio under the pea-protein-plus-inulin diet. In addition, protein retention appeared to be similar or even improved when also considering muscle mass in old rats fed the pea protein-plus-inulin diet, despite a higher fecal excretion of nitrogen. Nitrogen balance tended to be better in the group fed pea protein-plus-inulin, showing that the presence of dietary fiber enabled more efficient utilization of dietary nitrogen.

One of the most remarkable results of the present work is the better maintenance of type-2 muscle mass in old rats fed dietary fiber in addition to pea protein compared to rats fed only pea protein. The effects of plant proteins and inulin on muscle in old rats can vary depending on several factors. We previously reported that plant-source proteins, such as pea, lentil or faba bean, can have beneficial effects on muscle health in older rats [9,16]. In particular, we demonstrated that pea proteins had a similar effect on nitrogen balance, true digestibility, net protein utilization, body composition, tissue weight, skeletal muscle protein synthesis or degradation, and muscle mitochondrial activity in old rats as compared to a control group [16]. In addition, the inclusion of peas as a source of protein in the diets of growing rats was shown to lead to similar fractional synthesis rates of liver and muscle proteins to casein-fed controls [36].

The positive effect on muscle of pea protein consumed with inulin in old rats may also be partially fiber-dependent. Inulin is a type of dietary fiber that is widely found in many plants. Inulin acts as a prebiotic, meaning it serves as food for beneficial gut bacteria [30]. While there is limited direct research on inulin and muscle health in old rats, a healthy gut microbiota composition is associated with improved nutrient absorption, reduced inflammation, and enhanced overall health, all of which indirectly supports muscle health. In addition, inulin consumption has been linked to improved glucose metabolism and insulin sensitivity [37]. Better metabolic control may also indirectly benefit muscle health, as insulin resistance and metabolic dysfunction can impair muscle function and disrupt muscle protein turnover [38]. Here we showed that the consumption of inulin can decrease certain markers of muscle proteolysis. Note that Wang et al. [39] previously showed that inulin supplementation in pigs significantly decreased the expression level of muscle-specific ubiquitin ligase MuRF-1, i.e., the same marker of proteolysis downregulated here. Moreover, although the ingestion of dietary fiber had no specific additional effect on muscle protein synthesis in old rats, there was a significant postprandial stimulation of muscle protein synthesis in these animals after the consumption of pea proteins. As already mentioned, the specific effect of plant proteins such as pea proteins on muscle protein synthesis has already been demonstrated in rodents, particularly in old rats [9,16]. Here, in the pea protein-plus-inulin group, we posit that a positive effect of pea protein on muscle protein synthesis rate coupled with a positive action of inulin on muscle proteolysis may lead to increased protein anabolism and thus contribute to maintenance of muscle mass. In addition, apart from a well-balanced amino acid content, pea protein can be accompanied by plant bioactives, such as polyphenols, which have an antioxidant action. It has previously been shown that antioxidants can stimulate muscle protein synthesis in old rats [10].

Dietary fibers that are not digested by host enzymes but fermented by gut bacteria could modulate the gut microbiome composition within a relatively short space of time. These fermented non-digestible compounds favor the proliferation of health-promoting bacteria that may positively affect muscle health. In older adults affected by frailty syndrome, the administration of a prebiotic significantly improved muscle-related frailty criteria, e.g., exhaustion and handgrip strength [40], compared with a placebo. Cani et al. reported decreased levels of inflammation and increased muscle mass in obese mice supplemented with oligofructose fiber [41], and the beneficial effect of prebiotic administration on gut microbiota was further confirmed by increased levels of *Lactobacillus* and *Bifidobacterium* spp. found in follow-up analysis [42]. Recently, Giron et al. showed that when one of these bacteria, *Lactobacillus*, was ingested daily for 1 month, food-restricted 18-month-old rats were able to preserve muscle protein mass by improving both muscle protein synthesis and insulin sensitivity [43]. These findings suggest that *Lactobacillus* and *Bifidobacterium* may influence gut–muscle communication and regulate muscle size. Intestinal *Bifidobacterium* content decreases with age [44], and an age-related decrease in gut *Bifidobacterium* content may be the mechanism underpinning the increase in circulating endotoxin that induces muscle atrophy [45]. Interestingly, supplementation with galactooligosaccharides (GOS) in middle-aged and older people was shown to attenuate the age-associated reduction in gut *Bifidobacteria* [46]. In particular, the GOS treatment led to an increase in the number of *Bifidobacteria* and *Lactobacilli* together with higher butyrate levels. In the same way, inulin enhances the growth of indigenous *Lactobacilli* and *Bifidobacteria* by inducing colonic production of short-chain fatty acids (SCFA), and these properties are related to decreased mucosal lesion scores and diminished mucosal inflammation [15]. Thus, inulin, by increasing the number of beneficial bacteria on the mucosal surface, may improve the gut mucosal barrier and prevent gastrointestinal infections with enteric pathogens as well as systemic inflammation from the translocation of gut bacteria and inflammatory by-products [45]. Inulin has shown anti-inflammatory properties by decreasing pro-inflammatory markers [47]. As chronic inflammation can negatively impact muscle health and is associated with age-related muscle decline [48], reducing inflammation may support muscle function and prevent age-related muscle decline. The administration of a symbiotic comprising the probiotic *Bifidobacterium longum* and an inulin-based prebiotic component has also been demonstrated to enhance butyrate production and blunt proinflammatory responses [49]. This and other studies based on inulin supplementation converge towards the hypothesis that administration of the pea protein-plus-inulin mixture in advanced age might positively affect the microbiota and slow the age-associated decline of muscle mass and function. Taken together, these lines of evidence support the idea that pre- and/or probiotic supplementation may prevent age-related muscle loss by increasing the abundance of *Bifidobacterium* and butyrate producers in old individuals. In addition, plant proteins, especially those containing some remaining bioactive compounds such as antioxidants and polyphenols, may help reduce inflammation and further support muscle health.

Our work also shows that an additional dietary fiber intake in addition to plant proteins improved muscle mitochondrial function in old rats compared to a pea protein intake alone. The direct effects of dietary fiber on mitochondria in muscle have not been extensively studied, but their indirect impact on muscle health and performance suggests potential benefits for mitochondrial function. Emerging evidence suggests that the gut microbiota may affect mitochondrial function in several tissues [50,51,52], including skeletal muscle [53]. Gut dysbiosis increases the permeability of intestinal mucosa, promotes systemic inflammation, subclinical immune activation and insulin resistance, and ultimately leads to muscle mitochondria damage [54]. Studies have demonstrated that probiotic intake or fecal microbial transplantation can regulate mitochondrial energy metabolism and skeletal muscle functions in advanced age [55,56]. For example, Chen et al. showed that *Lactobacillus casei* supplementation in old mice produces anti-inflammatory effects and leads to maintained muscle mitochondrial functions [57]. Very recently, Zhang et al. reported that insoluble dietary fiber intake in rats prevents obesity and improves the dyslipidemia and hepatic steatosis caused by a high-fat diet by promoting hepatic mitochondrial fatty acid oxidation [58]. Importantly, this intervention promotes medium- and long-chain fatty acid oxidation in hepatic mitochondria by improving the contents of key mitochondrial enzymes such as carnitine palmitoyl transferase-1, acyl-coenzyme A oxidase 1, acetyl coenzyme A synthase, and acetyl coenzyme A carboxylase [58]. Further studies are needed to explore the specific effect of dietary fiber on muscle mitochondria, but this action may also exist in skeletal muscle. Here we show that the consumption of inulin specifically increases the activity of enzymes that play a central role in the citric acid cycle, beta-oxidation, and the respiratory chain, i.e., complex 1, citrate synthase, and 3-hydroxyacyl-CoA dehydrogenase (HAD) activities, in plantaris muscle. Although the underlying mechanism needs to be studied further, the existing evidence shows that certain gut bacteria produce metabolites that can influence mitochondrial metabolism and energy production. Note that SCFA, such as butyrate, are key intermediates affecting the gut–muscle axis [59]. SCFA can be used for de novo synthesis of lipids and glucose, which are the main energy sources for the host. In addition, two orphan G-protein-coupled receptors, GPR41 and GPR43, were reported to be activated by SCFA. SCFA-mediated GPR43 activation suppressed insulin signaling in adipose tissue, leading to inhibition of fat accumulation [60]. The study found that unincorporated lipids and glucose were primarily utilized in muscles where the expression levels of energy expenditure, glycolysis and β-oxidation-related genes was increased [60]. In another study, the incorporation of butyrate in the diet induced higher energy expenditure and oxygen consumption in mice, suggesting an increase in fatty acid oxidation that was confirmed by monitoring 14C-labeled palmitic acid in butyrate-treated mice [61]. The authors explained the observed effects by an activation of peroxisome proliferator-activated receptor-γ coactivator-1α (PGC-1α), which is a master regulator of mitochondrial biogenesis, in brown fat, skeletal muscle and liver [61]. Interestingly, we also found that muscle PGC-1α was also upregulated here. Taken as a whole, SCFA are good candidates to explain the biological effects of dietary fiber consumption on muscle mitochondrial function found here, as they can serve as substrates for mitochondria and regulate key mitochondrial functions [58]. By supporting a diverse and balanced gut microbiota, dietary fiber may indirectly affect mitochondrial function in muscle. In addition, the chronic low-grade inflammation observed during aging, i.e., Inflamm-aging, can negatively impact mitochondrial function [62]. Dietary fibers, particularly those with anti-inflammatory properties such as inulin, can help reduce systemic inflammation [47]. By lowering inflammation levels, dietary fibers may also contribute to better mitochondrial function in muscle.

This study has potential limitations. It would have been interesting to evaluate the composition of the microbiota of rats, in particular of those having consumed inulin. This analysis would have made it possible to better understand the impact of changes in gut bacterial abundance on the effect of inulin on skeletal muscle. Of note, the modification of the gut microbiota induced by a diet enriched with inulin has already been described in rats in different conditions, e.g., standard condition, obesity [63,64]. Inulin is usually regarded as a type of prebiotic, favorably stimulating the growth of *Bifidobacteria lactobacilli* and actinobacteria and inhibiting the growth of bacteroidetes. In addition, in this work we were unable to measure inflammatory markers for technical reasons. Inulin intake has been shown to reduce inflammation in various situations [15,45,47]. Age-related muscle loss is partly explained by an increased inflammatory state, i.e., increased production of pro-inflammatory cytokines. It therefore remains to be determined whether the intake of inulin in older individuals is able to reduce muscle loss by limiting inflammation.

In conclusion, pea protein is a plant-based source of high-quality protein that includes all the essential amino acids required to sustain muscle protein turnover. Furthermore, pea protein is generally well-tolerated and digestible, making it a suitable protein source for older people. Inulin is a type of dietary fiber found in various plants and is considered a prebiotic, which means that it provides nourishment to beneficial gut bacteria. The direct effects of inulin on muscle in old rats have been under-researched, but the evidence suggests it can promote a healthy gut microbiota by stimulating the growth of beneficial bacteria, and thus stave off systemic inflammation. Inulin has also been shown to improve glucose metabolism and insulin sensitivity. Dietary fiber intakes are generally too low in the general population and especially in older people, which may contribute to the main metabolic syndromes that accompany aging, such as sarcopenia. Note that caution is warranted when attempting to translate these findings from rats to humans, as there may be species-specific differences. Nonetheless, these findings provide promising insights into the potential benefits of combining pea protein with inulin for muscle health in older people. While more research is needed to fully understand the additional or even synergistic effects of plant-derived compounds such as proteins and fibers on muscle health during aging, there is already enough evidence to recommend combining intake of a high-quality protein with a sufficient fiber intake to protect muscle mass and function, and thus extend the autonomy of older people.

## Figures and Tables

**Figure 1 nutrients-15-03766-f001:**
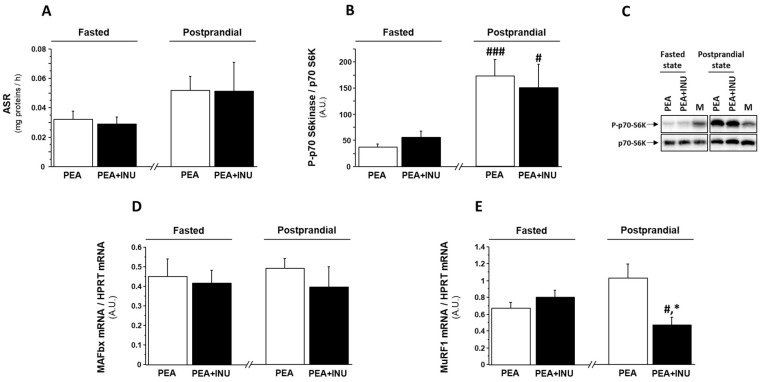
Protein synthesis rate, p70 S6 kinase phosphorylation rate, and expression of ubiquitin-proteasome pathway markers in plantaris muscles of fasted and postprandial old rats fed either PEA or PEA + INU diets. All rats were fasted overnight. PEA rats and PEA + INU rats were randomly divided into groups that either were kept in the fasted state or given a nutrient mixture by oral gavage (postprandial state). (**A**) Absolute synthesis rate was measured by tracer enrichment in plantaris muscles after incubation with L-[^13^C_6_] phenylalanine. (**B**,**C**) Ratio of phosphorylated p70 S6 kinase to total p70 S6 kinase content in the same plantaris muscles was determined by Western-blotting. Gene expressions of the two ubiquitin E3 ligases MAFbx (**D**) and MuRF1 (**E**) were assessed by quantitative RT-PCR. Data are expressed as means ± SEM. * significantly different from PEA rats at identical nutritional status (fasted or postprandial) at *p* < 0.05. # significantly different from the same rat group in the fasted state at *p* < 0.05. ### significantly different from the same rat group rats in the fasted state at *p* < 0.001. A.U.: arbitrary units. M: Mix of all samples. PEA rat group at the fasted state, n = 8; PEA rat group at the postprandial state, n = 6; PEA + INU rat group at the fasted state, n = 7; PEA + INU rat group at the postprandial state, n = 5.

**Figure 2 nutrients-15-03766-f002:**
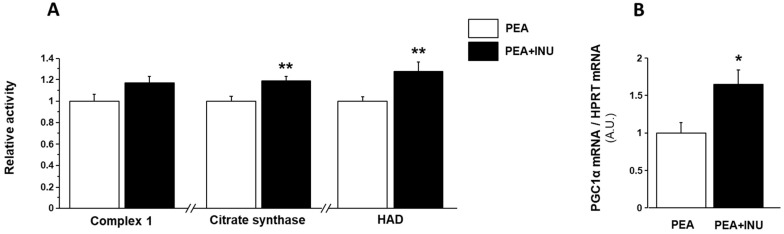
Mitochondrial function and biogenesis in plantaris muscles of PEA and PEA + INU old rats. (**A**) Mitochondrial function was assessed by measuring complex 1, citrate synthase, and 3-hydroxyacyl-CoA dehydrogenase (HAD) activity in plantaris muscles. Enzyme activities were expressed as fold change vs. the value found for the PEA rat group. (**B**) Mitochondrial biogenesis was determined by measuring PGC1α mRNA expression using quantitative RT-PCR and was expressed as fold change vs. value found for the PEA rat group. Data are expressed as means ± SEM. * significantly different from PEA rats at *p* < 0.05. ** significantly different from PEA rats at *p* < 0.01. (A.U.): arbitrary units. PEA rat group, n = 14; PEA + INU rat group, n = 12.

**Table 1 nutrients-15-03766-t001:** Composition of the experimental diets.

	PEA	PEA + INU
**Diet composition (g/100 g)**		
Pea protein	14	14
Fat (soybean oil)	6	6
Carbohydrates	68	68
Cellulose	7.5	0
Inulin	0	7.5
Vitamin and mineral mix	4.5	4.5

**Table 2 nutrients-15-03766-t002:** Composition of the nutrient mixture used for oral gavage.

Component	Final Concentration or Volume
Leucine	16 g/L
Sucrose	191 g/L
Glucose	191 g/L
Glutamine	0.3 g/L
Amino acid solution (M5550–Sigma Aldrich)	2 mL
Distilled water	q.s. 50 mL

**Table 3 nutrients-15-03766-t003:** Primer sequences used for quantitative analysis of gene expression.

Gene Name	Forward and Reverse Primers
MAFbx Muscle atrophy F-box	*For 5*′-AGTGAAGACCGGCTACTGTGGAA-*3*′ *Rev 5*′-TTGCAAAGCTGCAGGGTGAC-*3*′
MuRF1 Muscle RING finger-1	*For 5*′-GTGAAGTTGCCCCCTTACAA-*3*′ *Rev 5*′-TGGAGATGCAATTGCTCAGT-*3*′
PGC1α Peroxisome proliferator-activated receptor gamma coactivator 1-alpha	*For 5*′-AGTTTTTGGTGAAATTGAGGAAT-*3*′ *Rev 5*′-TCATACTTGCTCTTGGTGGAAGC-*3*′
HPRT Hypoxanthine-guanine phosphoribosyltransferase	*For 5*′-AGTTGAGAGATCATCTCCAC-*3*′ *Rev 5*′-TTGCTGACCTGCTGGATTAC-*3*′

**Table 4 nutrients-15-03766-t004:** Body weight, fat mass and lean mass changes after 16 weeks of the experimental study.

	PEA (n = 14)	PEA + INU (n = 12)	*p*
**Body weight (g)**			
initial	598 ± 19	577 ± 18	0.4
final	612 ± 27	588 ± 21	0.5
change	+13 ± 16	+12 ± 12	0.9
**Fat mass (g)**			
initial	108 ± 8	89 ± 8	0.1
final	140 ± 16 ^#^	115 ± 12 ^##^	0.2
change	+33 ± 12	+26 ± 7	0.6
**Lean mass (g)**			
initial	438 ± 12	436 ± 13	0.9
final	414 ± 13 ^###^	418 ± 14 ^##^	0.8
change	−23 ± 4	−18 ± 5	0.4

Body weight change = (final body weight) minus (initial body weight). Fat mass change = (final fat mass) minus (initial fat mass). Lean mass change = (final lean mass) minus (initial lean mass). Results are expressed as means ± SEM. *p* indicates the *p*-value for the statistical analysis comparing PEA rats and PEA + INU rats. ^#^ significantly different from initial value with *p* < 0.05. ^##^ significantly different from initial value with *p* < 0.01. ^###^ significantly different from initial value with *p* < 0.001.

**Table 5 nutrients-15-03766-t005:** Relative tissue weights in PEA and PEA + INU rats after 16 weeks of the experimental study.

	PEA (n = 14)	PEA + INU (n = 12)
Plantaris (mg/100 g body weight)	46.0 ± 1.6	53.5 ± 3.4 *
Tibialis (mg/100 g body weight)	70.6 ± 2.7	82.0 ± 4.1 *
Gastrocnemius (mg/100 g body weight)	202.1 ± 9.0	238.9 ± 15.4 *
Soleus (mg/100 g body weight)	29.8 ± 1.9	33.9 ± 2.9
Perirenal adipose tissue (g/100 g body weight)	3.0 ± 0.3	2.2 ± 0.2
Subcutaneous adipose tissue (g/100 g body weight)	2.3 ± 0.2	1.9 ± 0.3
Liver (g/100 g body weight)	2.3 ± 0.1	2.2 ± 0.1
Heart (g/100 g body weight)	0.3 ± 0.0	0.3 ± 0.0

Results are expressed as means ± SEM. * significantly different from PEA rats at *p* < 0.05.

**Table 6 nutrients-15-03766-t006:** Evaluation of protein quality during the 4-day period in metabolic cages.

	PEA (n = 14)	PEA + INU (n = 12)
Nitrogen intake (g)	1.65 ± 0.09	1.86 ± 0.11
Fecal nitrogen (g)	0.13 ± 0.01	0.18 ± 0.01 *
Urinary nitrogen (g)	0.92 ± 0.07	0.86 ± 0.06
FN/NI	0.08 ± 0.01	0.10 ± 0.01 *
UN/NI	0.56 ± 0.03	0.47 ± 0.03
Nitrogen balance (g)	0.60 ± 0.07	0.81 ± 0.09

NI = Nitrogen Intake. FN = Fecal Nitrogen. UN = Urinary Nitrogen. Results are expressed as means ± SEM. * significantly different from PEA rats at *p* < 0.05.

**Table 7 nutrients-15-03766-t007:** Metabolic parameters in plasma of PEA and PEA + INU rats in fasted vs. postprandial state.

	PEA		PEA + INU	
	Fasted State (n = 8)	Postprandial State (n = 6)	Fasted State (n = 7)	Postprandial State (n = 5)
Glucose (g/L)	0.970 ± 0.108	1.574 ± 0.072 ***	0.795 ± 0.112	1.445 ± 0.083 ***
Insulin (ng/mL)	0.553 ± 0.102	2.531 ± 0.456 ***	0.564 ± 0.125	2.037 ± 1.066 *
Triglycerides (g/L)	0.604 ± 0.176	0.745 ± 0.133	0.682 ± 0.139	0.486 ± 0.036

Results are expressed as means ± SEM. * Mean values significantly different between fasted rats and postprandial rats fed with the same diet at *p* < 0.05. *** Mean values significantly different between fasted rats and postprandial rats fed with the same diet at *p* < 0.001.

## Data Availability

Not applicable.

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
