# Peer review of "Associating Inulin with a Pea Protein Improves Fast-Twitch Skeletal Muscle Mass and Muscle Mitochondrial Activities in Old Rats"

_nutrients, 2023, doi:10.3390/nu15173766_

Round 1

Reviewer 1 Report

The paper of Salles et al. describes the effect of pea proteins given with inulin on skeletal muscle in old rats.  Although the paper is rich of data which are well performed and presented, there are some aspects not clear.

  1. Was there a control group of rats?
  2. Which was used as an internal control in the Western blot analysis? Is it possible to show a representative blot?
  3. In my opinion representative blots for phospho p70 S6 kinase and total p70 S6 kinase should be shown.

Author Response

We are grateful to the reviewers for their valuable suggestions for improving the manuscript.

The paper of Salles et al. describes the effect of pea proteins given with inulin on skeletal muscle in old rats. Although the paper is rich of data which are well performed and presented, there are some aspects not clear.

  1. Was there a control group of rats?

The present article follows another work we published in Nutrients a few months ago (1). To sum-up, in this study, thirty 20-month-old male Wistar rats were assigned to a control diet (control group) and to an isoproteic and isocaloric diet containing pea protein isolate for 16 weeks. The pea protein was exactly the same (same batch) as that used in the present work. Furthermore, the treatment duration with this protein was also the same. The animals were the same age and came from the same supplier (thus had been raised under the same conditions). Feeding the rats with the control diet or the diet containing the pea protein induced a similar effect on nitrogen balance, true digestibility and net protein utilization in old rats. The two different protein sources did not alter body composition, tissue weight, skeletal muscle protein synthesis or degradation. Muscle mitochondrial activity, inflammation status and insulin resistance were similar between the two groups. Therefore, old rats used pea protein with the same efficiency as casein, i.e. the protein of the control diet, due to its high digestibility and amino acid composition.

Due to the results of this first work and in order to reduce the number of animals used (as requested by our ethics committee), we decided not to repeat the study comparing old rats fed with a diet containing pea protein and rats fed a control diet.

In addition the aim of the present work was to compare the effect of plant proteins, i.e. pea proteins, to the effect of plant proteins, i.e. pea proteins, given with dietary fibers. We thought that a fiber intake could potentiate the effect of dietary proteins in the context of a « more green diet » in older people. Also the comparison between a control group and the group consuming pea protein and fiber did not seem relevant to us.

We have added a sentence in the discussion section to make it clear to the reader that we have already explored the effect of pea protein versus a control diet in old rats (see page 8, lines 359-362).

1.Salles J, Guillet C, Le Bacquer O, Malnero-Fernandez C, Giraudet C, Patrac V, et al. Pea Proteins Have Anabolic Effects Comparable to Milk Proteins on Whole Body Protein Retention and Muscle Protein Metabolism in Old Rats. Nutrients. 2021;13(12).

  1. Which was used as an internal control in the Western blot analysis? Is it possible to show a representative blot?
  2. In my opinion representative blots for phospho p70 S6 kinase and total p70 S6 kinase should be shown.

The internal control used to normalize the blots among each other was a mixture of all the samples and was consistently applied to each gel. Unfortunately, we are unable to display representative blots of the p70 s6 kinase quantification without cropping and editing the images. Indeed, the differences of phosphorylation of p70-S6 kinase in the plantaris muscle were very strong between the fasting state and the postprandial state. Therefore, we decided to run the samples from fasted rats and postprandial rats on separate gels. A mixture of all samples was loaded on each gel to allow for comparison of the blots. We included two representative blots in Figure 1B (See below and page 14). We added this information in the Materials and Methods section (see page 5, lines 217-221).

Reviewer 2 Report

Comments:

Review of the paper

nutrients-2501965

This study aimed to investigate the effect of pea proteins given with inulin on skeletal muscle in old rats. There are two diet groups of rats: a ‘PEA’ group receiving a pea protein-based diet, or a ‘PEA+INU’ group receiving the same pea protein-based diet supplemented with inulin. Through comparison of weight, muscle p70 S6 kinase phosphorylation rate, evaluation of protein quality, MuRF1 transcript levels, PGC1-α expression and other indicators, it is clarified that dietary supplementation with pea proteins in combination with inulin has the potential to attenuate age-related muscle loss. Although it is somewhat innovative, the data is not convincing enough. The section specific comments are listed below.

MATERIALS AND METHODS

-The author needs to explain the basis for selecting the measured indicators.

-In this study, 40 rats participated in the experiment at the beginning, but by the end of the experiment, only 14 rats in the PEA group and 12 rats in the PEA+INU group underwent further analysis. Sample loss may affect the reliability of the results. In addition, the specific reasons and handling methods for sample loss were not provided in the original text.

RESULTS

-The basic physical measurement data of the rats involved in this experiment need to be supplement.

-In Table 4 ,it’s better to mark the “p”.

-In addition to the comparison between the two groups, it also needs to be compared before and after the experiment.

DISCUSSION

- In the discussion, it is mentioned many times that the inulin on the gut microbiota, is it possible to increase the analysis of feces to better understand its impact on intestinal flora?

-As mentioned in the results, these functions of inulin have the potential to attenuate age-related 28 muscle loss. What about normal-age rats?

Minor editing of English language required

Author Response

We are grateful to the reviewers for their valuable suggestions for improving the manuscript.

This study aimed to investigate the effect of pea proteins given with inulin on skeletal muscle in old rats. There are two dietgroups of rats: a ‘PEA’ group receiving a pea protein-based diet, or a ‘PEA+INU’ group receiving the same pea protein-based diet supplemented with inulin. Through comparison of weight, muscle p70 S6 kinase phosphorylation rate, evaluation of protein quality, MuRF1 transcript levels, PGC1-αexpression and other indicators, it is clarified that dietary supplementation with pea proteins in combination with inulin has the potential to attenuate age-related muscle loss. Although it is somewhat innovative, the data is not convincing enough. The section specific comments are listed below.

MATERIALS AND METHODS

-The author needs to explain the basis for selecting the measured indicators.

A sentence explaining the choice of indicators has been added for each indicator. See the Materials and Methods Section: page3, lines 146-147 ; page 4, lines 151-152, lines 161-162, lines 167-168, lines 186-188; page 5 lines 203-204, lines 224-225.

-In this study, 40 rats participated in the experiment at the beginning, but by the end of the experiment, only 14 rats in the PEA group and 12 rats in the PEA+INU group underwent further analysis. Sample loss may affect the reliability of the results. In addition, the specific reasons and handling methods for sample loss were not provided in the original text.

As we mentioned in the Materials and methods section, we purchased forty 20-month-old male Wistar rats from Janvier. Several animals died during the experiment or developed tumors. We excluded them from all the statistical analyses as indicated in the Statistics section. Hence, at the end of the experiment, there were 26 remaining animals as indicated in the table below.

Group

Remaining rats at the end of protocol

PEA

14

PEA+INU

12

Note that we observed a comparable mortality rate between both rat groups and that we did not perform any autopsies on spontaneously died animals. This is a typical mortality rate for Wistar rats aged 20-24 months. We anticipate this mortality by initiating our protocols with more rats than necessary for the analyses.

RESULTS

-The basic physical measurement data of the rats involved in this experiment need to be supplement.

-In Table 4 ,it’s better to mark the “p”.

-In addition to the comparison between the two groups, it also needs to be compared before and after the experiment.

As suggested by the reviewer, we added all these data in the table 4 (see below and pages 17-18 of the manuscript).

Table 4. Body weight, fat mass and lean mass changes after 16 weeks of the experimental study.

PEA

PEA+INU

p

Body weight (g)

initial

598 ± 19

577 ± 18

0.4

final

612 ± 27

588 ± 21

0.5

change

+13 ± 16

+12 ± 12

0.9

Fat mass (g)

initial

108 ± 8

89 ± 8

0.1

final

140 ± 16 #

115 ± 12 ##

0.2

change

+33 ± 12

+26 ± 7

0.6

Lean mass (g)

initial

438 ± 12

436 ± 13

0.9

final

414 ± 13 ###

418 ± 14 ##

0.8

change

-23 ± 4

-18 ± 5

0.4

Body weight change = (Final body weight) minus (initial body weight)

Fat mass change = (Final fat mass) minus (initial fat mass)

Lean mass change = (Final lean mass) minus (initial lean mass)

Results are expressed as means ± SEM.

p indicates the p-value for the statistical analysis comparing PEA rats and PEA+INU rats.

# significantly different from initial value with p<0.05.

## significantly different from initial value with p<0.01.

### significantly different from initial value with p<0.001.

We modified these sentences in Results section (page 6, lines 253-256):

“At the beginning and at the end of the study, there was no between-group difference in body weight, fat mass and lean mass. PEA and PEA+INU rats tended to gain body weight, significantly gained fat mass and significantly lost lean mass over the course of the 16-week experimental period, and these changes were similar between groups (Table 4).”

We added this sentence in the Materials and methods section (Statistics, page 5, lines 244-246):

“To compare initial values of body weight, fat mass and lean mass with final values, data were analyzed by a paired t test.”

DISCUSSION

- In the discussion, it is mentioned many times that the inulin on the gut microbiota, is it possible to increase the analysis of feces to better understand its impact on intestinal flora?

The reviewer is right, it would have been interesting to measure the composition of the gut microbiota and its modification with the intake of inulin. However, for technical and financial reasons, we were unable to carry out such analyses.

Nevertheless, the modification of the gut microbiota induced by a diet enriched with inulin has already been described in rats in different conditions, e.g. standard condition, obesity (1, ). Inulin is usually regarded as a type of prebiotic, favorably stimulating the growth of bifidobacteria and lactobacilli in rodents. It was reported that the abundance of Actinobacteria significantly increased and that of Bacteroidetes decreased in rats after inulin-containing diet. In addition, Enterococcus was promoted and levels of Bifidobacterium and Olsenella both increased after inulin diet. In these studies, one of the most striking observation was that Olsenella became a dominant genus comparable with Bifidobacterium after inulin intake. The capacity of some specific strains of these genus to use inulin was also described.

As we did not do again such an analysis of the intestinal microbiota, we added a chapter describing this limitation. See the Discussion Section, page 10, lines 480-492.

  1. Mao B, Li D, Zhao J, Liu X, Gu Z, Chen YQ, et al. Metagenomic insights into the effects of fructo-oligosaccharides (FOS) on the composition of fecal microbiota in mice. J Agric Food Chem. 2015;63(3):856-63.
  2. Miralles-Perez B, Nogues MR, Sanchez-Martos V, Fortuno-Mar A, Ramos-Romero S, Torres JL, et al. Influence of Dietary Inulin on Fecal Microbiota, Cardiometabolic Risk Factors, Eicosanoids, and Oxidative Stress in Rats Fed a High-Fat Diet. Foods. 2022;11(24).

-As mentioned in the results, these functions of inulin have the potential to attenuate age-related 28 muscle loss. What about normal-age rats?

To our knowledge, a study dealing with the effect of inulin on muscle mass in young rat has not yet been carried out. Similarly, its effect in situations inducing muscle loss, such as chronic diseases, has not yet been done.

Reviewer 3 Report

The study addresses a fascinating subject that could be a strategy to improve skeletal muscle mass and muscle mitochondrial activity in aging. However. I made some considerations for the authors to improve the manuscript.

1. Why did the authors only use male rats? Wouldn’t it be interesting to evaluate in females, since menopause causes harmful effects on musculature?

2. I think the authors could talk a little more about inulin in the introduction since it is part of the aim of the study and the authors only talked about an inulin precursor.

3. “Here we hypothesized that consumption of a prebiotic such as inulin in addition to a plant-based diet rich in pea protein would increase the effect of the plant protein on muscle mass in old rats and that this effect would be associated with profound changes in muscle protein and energy metabolism”. Are there studies in the literature showing that the prebiotic with inulin potentiates the action of a plant-based diet in the muscle? The authors only mentioned that the prebiotic with 1-kestose, a precursor of inulin, increased muscle mass. Therefore, inulin alone already promotes this effect, what would be the purpose of adding plant-based protein?

4. “At the end of the experiment, remaining rats (PEA rats; n=14 and PEA+INU; rats 124 n=12) were fasted overnight but with free access to water”. There were 20 rats in each group, what happened to the animals?

5. The authors analyzed by western blot p70 S6 kinase, please add the bands so that they are represented together with the graph.

6. Page 7, lines 306-308: “Pea protein is a good source of essential amino acids, which are crucial for muscle protein turnover. Inulin, acting as a prebiotic, can positively influence gut microbiota, leading to reduced inflammation and metabolic improvements”. Please add a reference. Why did the authors not assess these mice's microbiota and inflammatory status?

7. Add the number of animals in each group to the figure captions.

8. Table 7: Why did the authors not compare the glycemia of the animals in the PEA group with the animals in the PEA+INU group?

9. I don´t know if the title of the study reflects the aim, the idea that the study gives me is that associating inulin with a diet pea brings beneficial effects on the muscle, and the title is saying that inulin improves muscle contraction and mitochondrial activity in fed old rats with pea protein diet. I think the authors need to appreciate that the plant-based diet also has benefits and in itself has a significant effect on muscle. In addition, I missed a group with a normal diet without pea protein and without inulin pea protein to show the effect of pea protein and pea protein+inulin.

Author Response

We are grateful to the reviewers for their valuable suggestions for improving the manuscript.

The study addresses a fascinating subject that could be a strategy to improve skeletal muscle mass and muscle mitochondrial activity in aging. However. I made some considerations for the authors to improve the manuscript.

  1. Why did the authors only use male rats? Wouldn’t it be interesting to evaluate in females, since menopause causes harmful effects on musculature?

Male rats are usually used in studies investigating the effect of age on muscle mass and function. We therefore chose to use male rats in order to be able to interpret and compare our data with regard to that of the literature. In addition, we have published a first work focusing on the effects of dietary pea protein versus milk protein on muscle in old rats (1). We would like to extend this first study with work focusing on the interest of dietary fiber when consuming the same plant-protein, i.e. pea protein.

Nevertheless, the reviewer is right, these studies will have to be done again with female rats in order to evaluate the sex-related change of muscle mass to pea proteins and inulin intake. As the female rats do not undergo menopause, i.e. menopause does not exist in rat, it will be necessary to do ovaries excision in order to obtain a model of aging approaching that observed in Humans. This surgery will lead to an experimental bias compared to non-operated male rats. Also, the rodent model does not seem to be the most suitable for evaluating the effect of age on muscle mass and metabolism in male and female individuals.

  1. Salles J, Guillet C, Le Bacquer O, Malnero-Fernandez C, Giraudet C, Patrac V, et al. Pea Proteins Have Anabolic Effects Comparable to Milk Proteins on Whole Body Protein Retention and Muscle Protein Metabolism in Old Rats. Nutrients. 2021;13(12).

  1. I think the authors could talk a little more about inulin in the introduction since it is part of the aim of the study and the authors only talked about an inulin precursor.

A chapter has been added in the introduction section to explain what inulin is. See page 2 lines 90-95 in the introduction section.

  1. “Here we hypothesized that consumption of a prebiotic such as inulin in addition to a plant-based diet rich in pea protein would increase the effect of the plant protein on muscle mass in old rats and that this effect would be associated with profound changes in muscle protein and energy metabolism”. Are there studies in the literature showing that the prebiotic with inulin potentiates the action of a plant-based diet in the muscle? The authors only mentioned that the prebiotic with 1-kestose, a precursor of inulin, increased muscle mass. Therefore, inulin alone already promotes this effect, what would be the purpose of adding plant-based protein?

1-kestose intake appears to improve muscle mass in sarcopenic men. There is no accurate dietary survey of these individuals, especially for proteins. Current nutritional and ecological recommendations require us to make our diet more plant-based, especially with regard to dietary protein. As a result, the use of dietary plant protein is currently developing in the population with the emergence of new corresponding products on the shelves. In addition, it is well known that the older people decrease their protein intake from animal sources for various reasons, e.g. loss of appetite for these kinds of foods, teething problems, economic issues...

Therefore, the evaluation of the effects of different components of plant foods on muscle mass and function is now necessary. We started this work by comparing the effectiveness of plant proteins compared to animal proteins on muscle mass and function. We would like to go further by evaluating the impact of dietary fibre, which can be provided in the same plant food as protein, on muscle in old rats. Also, we bring new clues concerning the overall interest of plant foods, proteins and other nutrients composing them, on muscle health during aging.

  1. “At the end of the experiment, remaining rats (PEA rats; n=14 and PEA+INU; rats 124 n=12) were fasted overnight but with free access to water”. There were 20 rats in each group, what happened to the animals?

As we mentioned in the Materials and methods section, we purchased forty 20-month-old male Wistar rats from Janvier. Several animals died during the experiment or developed tumors. We excluded them from all the statistical analyses as indicated in the Statistics section. Hence, at the end of the experiment, there were 26 remaining animals as indicated in the table below.

Group

Remaining rats at the end of protocol

PEA

14

PEA+INU

12

Note that we observed a comparable mortality rate between both rat groups and that we did not perform any autopsies on spontaneously died animals. This is a typical mortality rate for Wistar rats aged 20-24 months. We anticipate this mortality by initiating our protocols with more rats than necessary for the analyses.

  1. The authors analyzed by western blot p70 S6 kinase, please add the bands so that they are represented together with the graph.

Unfortunately, we are unable to display representative blots of the p70 s6 kinase quantification without cropping and editing the images. Indeed, the differences of phosphorylation of p70-S6 kinase in the plantaris muscle were very strong between the fasting state and the postprandial state. Therefore, we decided to run the samples from fasted rats and postprandial rats on separate gels. A mixture of all samples used as an internal control was loaded on each gel to allow for comparison of the blots. We included two representative blots in Figure 1B (See below and page 14). We added this information in the Materials and Methods section (see page 5, lines 217-221).

  1. Page 7, lines 306-308: “Pea protein is a good source of essential amino acids, which are crucial for muscle protein turnover. Inulin, acting as a prebiotic, can positively influence gut microbiota, leading to reduced inflammation and metabolic improvements”. Please add a reference. Why did the authors not assess these mice's microbiota and inflammatory status?

The reviewer is right, it would have been interesting to measure the composition of the gut microbiota and its modification with the intake of inulin. However, for technical and financial reasons, we were unable to carry out such analyses.

Nevertheless, the modification of the gut microbiota induced by a diet enriched with inulin has already been described in rats in different conditions, e.g. standard condition, obesity (2, 3). Inulin is usually regarded as a type of prebiotic, favorably stimulating the growth of bifidobacteria and lactobacilli in rodents. It was reported that the abundance of Actinobacteria significantly increased and that of Bacteroidetes decreased in rats after inulin-containing diet. In addition, Enterococcus was promoted and levels of Bifidobacterium and Olsenella both increased after inulin diet. In these studies, one of the most striking observation was that Olsenella became a dominant genus comparable with Bifidobacterium after inulin intake. The capacity of some specific strains of these genus to use inulin was also described.

As we did not do again such an analysis of the intestinal microbiota, we added a chapter describing this limitation. See the Discussion Section.

It would also have been interesting to measure inflammatory markers. For technical reasons, we were only able to measure the blood concentration of CRP (C-reactive protein). Blood CRP was not different between the two groups. However, we do not think that these are sufficient to conclude that inflammatory status is not changed. We have therefore chosen not to present this parameter.

We have added this comment in the limitation chapter of the discussion section. See the Discussion Section, page 10, lines 480-492.

  1. Mao B, Li D, Zhao J, Liu X, Gu Z, Chen YQ, et al. Metagenomic insights into the effects of fructo-oligosaccharides (FOS) on the composition of fecal microbiota in mice. J Agric Food Chem. 2015;63(3):856-63.
  2. Miralles-Perez B, Nogues MR, Sanchez-Martos V, Fortuno-Mar A, Ramos-Romero S, Torres JL, et al. Influence of Dietary Inulin on Fecal Microbiota, Cardiometabolic Risk Factors, Eicosanoids, and Oxidative Stress in Rats Fed a High-Fat Diet. Foods. 2022;11(24).

  1. Add the number of animals in each group to the figure captions.

As suggested by the reviewer, we added this information in each table and each figure legend.

  1. Table 7: Why did the authors not compare the glycemia of the animals in the PEA group with the animals in the PEA+INU group?

We apologize for not being specific enough. In the first version of the manuscript, a sentence indicated that “ … plasma levels of glucose, insulin and triglycerides were similar in both PEA and PEA+INU rats in both fasted state and postprandial state (Table 7).”

To be more precise, we modified the sentence as follows (Results section, page 6, lines 281-283):

“Note, that plasma levels of glucose, insulin and triglycerides were not significantly different in both PEA and PEA+INU rats in both fasted state and postprandial state (Table 7).”

  1. I don´t know if the title of the study reflects the aim, the idea that the study gives me is that associating inulin with a diet pea brings beneficial effects on the muscle, and the title is saying that inulin improves muscle contraction and mitochondrial activity in fed old rats with pea protein diet. I think the authors need to appreciate that the plant-based diet also has benefits and in itself has a significant effect on muscle. In addition, I missed a group with a normal diet without pea protein and without inulin pea protein to show the effect of pea protein and pea protein+inulin.

The reviewer is right, a plant-based diet already displays its own effects on muscle. Nevertheless, in the present work we have used pea proteins and not whole peas, e.g. pea flour. We recently published that pea protein has the same effect on muscle as animal dietary proteins such as casein. The mixture between pea protein and inulin potentiates this effect. We have changed the title to reflect the potential beneficial synergic effect of these two ingredients as follows:

“Associating inulin with a pea protein improves fast-twitch skeletal muscle mass and muscle mitochondrial activities in old rats.”

Concerning the control group, the present article follows another work we published in Nutrients a few months ago (1). To sum-up, in this study, thirty 20-month-old male Wistar rats were assigned to a control diet (control group) and to an isoproteic and isocaloric diet containing pea protein isolate for 16 weeks. The pea protein was exactly the same (same batch) as that used in the present work. Furthermore, the treatment duration with this protein was also the same. The animals were the same age and came from the same supplier (thus had been raised under the same conditions). Feeding the rats the control diet or the diet containing the pea protein induced a similar effect on nitrogen balance, true digestibility and net protein utilization in old rats. The two different protein sources did not alter body composition, tissue weight, skeletal muscle protein synthesis or degradation. Muscle mitochondrial activity, inflammation status and insulin resistance were similar between the two groups. Therefore, old rats used pea protein with the same efficiency as casein, i.e. the protein of the control diet, due to its high digestibility and amino acid composition.

Due to the results of this first work and in order to reduce the number of animals used (as requested by our ethics committee), we decided not to repeat the study comparing old rats fed with a diet containing pea protein and rats fed a control diet.

In addition the aim of the present work was to compare the effect of plant proteins, i.e. pea proteins, to the effect of plant proteins, i.e. pea proteins, given with dietary fibers. We thought that a fiber intake could potentiate the effect of dietary proteins in the context of a « more green diet » in older people. Also the comparison between a control group and the group consuming pea protein and fiber did not seem relevant to us.

  1. Salles J, Guillet C, Le Bacquer O, Malnero-Fernandez C, Giraudet C, Patrac V, et al. Pea Proteins Have Anabolic Effects Comparable to Milk Proteins on Whole Body Protein Retention and Muscle Protein Metabolism in Old Rats. Nutrients. 2021;13(12).

We have added a sentence in the discussion section to make it clear to the reader that we have already explored the effect of pea protein versus a control diet in old rats (see page 8, lines 359-362).

Round 2

Reviewer 3 Report

The authors adhered to the queries made by the reviewers by adding important information to the study.